# Multimodal Functional Imaging for Cancer/Tumor Microenvironments Based on MRI, EPRI, and PET

**DOI:** 10.3390/molecules26061614

**Published:** 2021-03-14

**Authors:** Ken-ichiro Matsumoto, James B. Mitchell, Murali C. Krishna

**Affiliations:** 1Quantitative RedOx Sensing Group, Department of Basic Medical Sciences for Radiation Damages, National Institute of Radiological Sciences, Quantum Medical Science Directorate, 4-9-1 Anagawa, Inage-ku, Chiba 263-8555, Japan; 2Radiation Biology Branch, Center for Cancer Research, National Cancer Institute, National Institutes of Health, Bethesda, MD 20892-1002, USA; jbm@helix.nih.gov

**Keywords:** theranostics, multimodal imaging, functional imaging, oxygen mapping, redox imaging, metabolic imaging

## Abstract

Radiation therapy is one of the main modalities to treat cancer/tumor. The response to radiation therapy, however, can be influenced by physiological and/or pathological conditions in the target tissues, especially by the low partial oxygen pressure and altered redox status in cancer/tumor tissues. Visualizing such cancer/tumor patho-physiological microenvironment would be a useful not only for planning radiotherapy but also to detect cancer/tumor in an earlier stage. Tumor hypoxia could be sensed by positron emission tomography (PET), electron paramagnetic resonance (EPR) oxygen mapping, and in vivo dynamic nuclear polarization (DNP) MRI. Tissue oxygenation could be visualized on a real-time basis by blood oxygen level dependent (BOLD) and/or tissue oxygen level dependent (TOLD) MRI signal. EPR imaging (EPRI) and/or T_1_-weighted MRI techniques can visualize tissue redox status non-invasively based on paramagnetic and diamagnetic conversions of nitroxyl radical contrast agent. ^13^C-DNP MRI can visualize glycometabolism of tumor/cancer tissues. Accurate co-registration of those multimodal images could make mechanisms of drug and/or relation of resulted biological effects clear. A multimodal instrument, such as PET-MRI, may have another possibility to link multiple functions. Functional imaging techniques individually developed to date have been converged on the concept of theranostics.

## 1. Introduction

Incidence of cancer/tumor is increased markedly with aging, and therefore the prevalence among elder people is high. Radiation therapy one of the three treatment modalities including surgery and chemotherapy in treating cancer/tumor. Ionizing radiation ionizes/excites water molecule and generates highly reactive species, i.e., free radical species and/or reactive oxygen species (ROS) [1,2,3]. Free radical species and/or ROS induced by water radiolysis can reach a target molecule through chain reactions mediated by membrane lipids, or form stable oxidizing species such as like H_2_O_2_. Free radicals and/or ROS generated oxidize biologically important molecules such as DNA causing single and double strand breaks [4,5]. Unrepaired DNA double strand breaks lead to cell death. Radiation therapy kills cancer cells by generation of free radicals in cancer cells and damaging DNA. This is so called indirect action of ionizing radiation. Though ionizing radiation can ionize the target molecule directly, the direct action for photon is relatively low (20–30%) compare to the indirect action (70–80%) [6,7,8]. Effects of radiation are essentially oxidative stress mediated by free radical species and/or ROS.

Tumor microenvironment is different compared to the normal tissues and has formed the basis for molecular imaging to detect and visualize these features [9,10]. Cancer/tumor tissues have hypoxia [11,12,13], low pH [14,15], higher glutathione concentrations [16,17], elevated aerobic glycolytic metabolism [18,19], etc. Hypoxia in the tumor tissue is due to immature vascular structure [13]. Energy production in such low pO_2_ environment in the cancer/tumor tissue may induce glycolytic activity of cancer/tumor cells, consequently causing a low pH environment as a result or lactate accumulation [20]. Visualizing such microenvironmental characteristics in tumors would be a useful not only for planning radiotherapy but also for early detection. Several medical imaging modalities emerged including magnetic resonance, nuclear medicine, and ultrasonic techniques to profile the tumor microenvironment.

Modern medical imaging techniques, such as X-ray computed tomography (CT), positron emission tomography (PET), and MRI play an important role in theranostics. The word theranostics suggests a technique or methodology with interface between the therapy and the diagnosis. Therapeutic monitoring is simultaneously done with diagnosis and is getting established as a new solid concept in a recent development of functional imaging techniques.

Quantifying hypoxia and/or redox status in cancer/tumor tissue is an important objective for theranostic medical imaging working with radiotherapy [21,22]. Radiation therapy can be tailored based on the tumor oxygen tension and/or redox status. A priori knowledge of tissue oxygen and/or redox status in the target and the surrounding tissues/organs will be helpful for planning a safe and efficient radiation therapy. In addition, visualizing metabolic changes in the cancer/tumor tissues can identify the target and degree of malignancy [23].

Analysis of biological information using MRI, electron paramagnetic resonance imaging (EPRI), and PET with a specific contrast agent have been developed to detect tumor microenvironment for achieving theranostic radiation therapy. In this review, we describe detection and visualization of tumor/cancer microenvironments, especially hypoxia, and the factors influenced by the hypoxic environment. Recent developments of translational multimodal imaging techniques using MRI, EPRI, and PET were introduced.

## 2. Historical Transitions of Modern Medical Imaging Techniques

Discovery of the X-ray made extraordinary contributions to the most fields of medical sciences. Rapid development of computer science after invention of integrated circuit (IC) has been absolutely imperative for the following development of medical imaging technology. The invention of X-ray CT [24,25] has drastically and innovatively changed medical image diagnostic systems. A clinical human X-ray CT was developed and commercialized simultaneously in 1973. Following the X-ray CT, the invention of MRI by Lauterbur [26] followed by the invention of pulsed MRI by Ernst and colleagues [27] revolutionized medical imaging. The first human MRI was commercialized in 1980. The main purpose of the CT and MRI at that time is observation of the anatomical information and the material changes of target tissue in a patient body non-invasively. The X-ray and MRI can give clear anatomical details inside the human body, especially hard issues by X-ray CT and soft tissues by MRI, respectively.

A more recent aspect of medical imaging technique is the molecular imaging, such as PET. PET imaging has been developed from early 1960. The PET scanner was commercialized in 1976 with the development of ^18^F-labeled deoxyglucose (^18^F-DG) [28]. The availability of ^18^F-DG-PET widely spread the concept of metabolic imaging. PET can map only distribution of positron-emitting radionuclide. Therefore, the PET image completely lacks anatomical information on it requiring a PET-CT system to provide anatomically co-registered metabolic scans. After more than 10 years from PET-CT, PET-MRI system was commercialized on 2015, and then the technical possibility of functional imaging has been expanded.

Electron paramagnetic resonance (EPR) is similar to nuclear magnetic resonance technique, and can provide images of the distribution paramagnetic species using similar strategies as in MRI with the use of magnetic field gradients for spatial encoding. Briefly EPR can detect paramagnetic species such as transition metal conplexes and free radicals. EPR imaging can detect only distribution of free radical species and therefore anatomical information is not available as is the case of PET. Development of EPR imaging also head toward molecular imaging such as redox imaging [29,30,31], which is a kind of dynamic imaging, and/or oxygen mapping [32,33,34], by application of spectral-spatial imaging [35,36,37].

After discovery of blood oxygen level dependent (BOLD) effect in 1990 [38,39,40], the term “functional MRI” (fMRI) is now synonymous for BOLD MRI, which will be described later. In next decades, MR spectroscopic imaging (MRSI) or also called chemical shift imaging has been actively investigated for cancer assessment and/or tumor tissue metabolism [41,42,43]. After the success of ^13^C hyperpolarized glycometabolic imaging [23], MRSI allowed imaging of enzymatic fluxes such as LDHA. Visualizing biological function has been the mainstream of recent developments of translational, preclinical, or clinical MRI techniques for early detection and grade estimation of disorders [44,45].

Multimodality is necessary for associating a function with anatomical information and for accurate diagnostics for tailored planning of therapeutics [21,46]. As described later, a specific contrast agent for each modality is necessary for visualizing a specific functional information, such as hypoxia, pO_2_, pH, redox status, and/or glycometabolisms. In addition, contrast agents having medicinal effects would be coming in [47,48,49].

## 3. Imaging Hypoxia by PET

Mapping hypoxia will be useful in planning of radiotherpay. PET imaging can map tumor hypoxia using positron-emission-nuclei-labeled contrast agent to image hypoxic environments [50]. Chemical structures of those PET probes are shown in Figure 1. The PET contrast agent is administered intravenously and its selective uptake in tissues is imaged using a PET scanner. [^64^Cu]Copper(II)-diacetyl-bis(N_4_-methylthiosemicarbazone (^64^Cu-ATSM) is a PET tracer used to map hypoxic tissue using PET [51]. Binding mechanism of ^64^Cu-ATSM into the hypoxic cell is probably irreversible and may not visualize dynamics of hypoxia in the tissue [52]. Whole body scans revealed uptake and retention of ^64^Cu-ATSM in the tumor-bearing leg, abdomen, and head region of the animals (Figure 2). [^18^F]fluoromisonidazole (^18^F-FMISO) is another hypoxic marker used in PET imaging. The tumor uptake of ^18^F-FMISO was clearly different from that of ^64^Cu-ATSM [52]. The tumor uptake of ^18^F-FMISO showed an increasing trend according with on oxygen content in breathing gas (10% oxygen > air > carbogen) in a mouse experiment, though no statistical difference was demonstrated. Predictable changes in tumor uptake of ^64^Cu-ATSM were unable to report on hypoxia when the oxygenation status of the tumor was modulated by breathing gas with the SCCVII tumor model in mouse [52]. In addition, ^64^Cu has 7 times long radioactive half-life (12.7 h), compared to the radioactive half-life of ^18^F (110 min). [^18^F]fluoroazomycin arabinoside (^18^F-FAZA), which is a ribose-nucleoside analog containing nitroimidazole ring in α-position of arabinose ring, hydrophilic hypoxia sensing contrast agent for PET with improved clearance and hypoxia targeting properties [53]. [^18^F]1-[2-fluoro-1-(hydroxymethyl)ethoxy]methyl-2-nitroimidazole (^18^F-FRP-170) showed uptake into viable hypoxic myocardium cells, and be expected to provide information for diagnosis of acute myocardial infarction [54]. Other hypoxic markers for PET, such as ^18^F-EF5 ([^18^F]-2-(2-nitroimidazol-1[*H*]-yl)-*N*-(2,2,3,3,3-pen-tafluoropropyl)-acetamide) [55] and ^18^F-HX4 (3-[^18^F]fluoro-2-(4-((2-nitro-1*H*-imidazol-1-yl)methyl)-1*H*-1,2,3-triazol-1-yl)propan-1-ol) [56], are developed to improve accuracy of hypoxic mapping on PET. Comparison of hypoxic markers for PET imaging is still in progress, most studies used ^18^F-FMISO as a comparable subject. An efficient hypoxic marker for PET image should be developed for the use of PET hypoxic mapping generally and widely.

Recently, a new concept of PET oximetry, which is estimating the life-span of positronium, was reported by Shibuya et al. [57]. Positronium is an unstable transitional state like atom with no nucleus consisting of positron and electron orbit the common center of mass. Some positrons released from positron-emission-nuclei capture an electron from surrounding molecules to form positronium. The life-span of the positronium is sensitive to coexisting molecular oxygen, i.e., paramagnetic species. Conversely, oxygen concentration in the sample can be estimated by measuring the lifetime of positronium. To measure the life-span of positronium, two signals, i.e., start and stop signals, of annihilation event are detected. The start signal is the positron emission, which can be determined by monitoring a prompt γ-ray using equipped Compton camera setting. The prompt γ-ray is emitted immediately after the positron emission from some isotopes, such as ^22^Na or ^44^Sc. ^44^Sc will be preferable label for biological/clinical use because of its shorter half-life (3.97 h) [58]. The stop signal is the positron annihilation, which can be determined by a pair of 511 keV photons. The method measures several ns time lag of the start and stop events. The instrumental configuration is a combination of PET and Compton gamma imaging [58]. A cylinder setup of Compton semiconducting detectors was arranged inside the cylinder of PET detectors. The prompt γ-ray is detected by a set of two detectors, inner Compton semiconducting detectors and outer usual scintillation photometer for PET, and direction of prompt γ-ray could be estimated as a solid angle based on Compton scattering angle estimation. The annihilation γ-ray is detected by a pair of outer scintillation photometers as usual PET detection, and linear direction of annihilation γ-ray could be estimated. Using both γ-ray scattering data and PET data, the coordinate of 2 intersection points on the surface of the cone and the line can be calculated. Final prediction of the image intensity would be converged by deselecting the data outside the view and accumulating data inside the view. This new technique can achieve a quantitative tissue pO_2_ imaging by PET equipped by Compton camera, when a biologically suitable molecular probe would be available.

The PET is a kind of auto-radiography using a positron-emission radio isotope nuclei, such as ^18^F or ^64^Cu, labeled chemical compound. The PET instrument detects a pair of 511 keV annihilation photons emitted by the positron-electron pair annihilation. The PET has big advantage to have multimodal detection of biological functions. What is sensing by PET is fundamentally depending on the molecular probe used. In other words, PET can detect not only for tissue glycodynamics or hypoxia but also have possibility to sense most of everything adapting on the biochemical reactions of positron-emission nuclei labeled molecular probes.

## 4. EPR Oxygen Mapping

The electron paramagnetic resonance (EPR) oxymetry technique is a non- or less-invasive and quantitative method for measuring oxygen concentration in a sample. The EPR oxymetry is based on measuring variation of relaxation time of electron spin on a paramagnetic probe. The paramagnetic probe can be dissolved in an aqueous solution or a solid paramagnetic probe can be implanted in a region of interest to monitor pO_2_ longitudinally. Molecular oxygen (O_2_) has two of unpaired electrons in its outermost orbitals. The two unpaired electrons on O_2_ cause EPR line broadening through shortening the relaxation time of electron spin of the infused or implanted paramagnetic probes through the spin-spin interaction. The pulsed EPR techniques can measure the T_1_, T_2_, and T_2_^*^ relaxation time of electron spin on the molecular probe directly. The relaxation time of electron spin is reflected on CW EPR linewidth. The relaxation time of electron spin was affected also on power saturation behavior on EPR signal intensity. By suitable calibration of the O_2_ induced line broadening, in vivo pO_2_ can be determined and imaged. EPR oxymetry techniques for in vivo oxygen mapping use several EPR based imaging modalities combined with an *i.v.* injectable nontoxic paramagnetic probe [59,60,61,62,63,64,65,66,67] including Overhauser MRI (OMRI) modality, which is also called as proton electron double resonance imaging (PEDRI) or dynamic nuclear polarization (DNP) [68,69,70]. Those EPR based methods, however, currently are limited only for experimental animals, such as mice or rats.

EPR oxygen mapping required a non-toxic and *i.v.* injectable paramagnetic probe, which should has narrow EPR linewidth as possible. For example, ^15^N-labeled nitroxides [33,34,70] and triarylmethyl radicals [61,62,63,64,65,66,67,68,69] have been used. Natural ^14^N nitroxides show triplet line EPR spectrum having relatively broad linewidth, while ^15^N-labeled nitroxides show narrower doublet line EPR spectrum (Figure 3A). Triarylmethyl radicals have narrow single line EPR spectrum (Figure 3B). Deuteration of the oxygen probe molecule makes the EPR linewidth narrower. Chemical structures of those oxygen probes were shown in Figure 3. Recently, triarylmethyl radical labeled molecular probe for probing membrane proteins, serum albumin are designed and reported [71,72,73].

EPR oximetry monitors effects of O_2_ on the relaxation time of stable free radical on the oxygen probe. The EPR linewidth is influenced by T_1_, T_2_, or T_2_* relaxation times. CW EPR spectral-spatial imaging (SSI) technique [33,34,61] reconstructs an image on a 3D (1D spectral and 2D spatial) or 4D (1D spectral and 3D spatial) matrix and then directly measures linewidth of reconstructed spectra (Figure 4). Pulsed EPR SSI working on single-point imaging (SPI) theory can measure T_2_* [62,63] (Figure 5). In addition, pulsed EPR SSI working on spin-echo theory can measure T_2_ [64], and repeating SSI or spin-echo correction with varying TR can estimate both T_1_ and T_2_* [65] or T_1_ and T_2_ [66]. Figure 6A shows an example of 3D oxygen mapping of a SCC tumor bearing mouse leg observed by SPI based 4D spectral-spatial imaging.

Varying the relaxation times can also shift the saturation state of microwave/radio frequency. Detecting signal loss by CW EPR power saturation can also mapping 3D O_2_ distribution [67]. This method is quite simple as a data acquisition process, which obtains just 2 images at 2 different RF power, i.e., non-saturated and enough-saturated conditions. Figure 6B shows an example of 3D oxygen mapping of a SCC tumor bearing mouse leg observed by the power saturation method.

Since EPR imaging can detect only the paramagnetic probe, therefore co-registration of EPR based oxygen map on the anatomical MRI observed on the corresponding position of subjected animal will be required to accurate distribution of hypoxia in the tissues [75,76]. Co-registration technique for several images of one identical subject observed by several different imaging modalities is necessary for achieving multimodal diagnosis and also described below. Another review paper introduced more details of EPR oxymetric imaging techniques and multimodal comparisons [74].

OMRI [68] or PEDRI [69], which is an instrument for observing double resonance of EPR and MRI, is detecting an enhancement of MRI signal through Overhauser effect, which is also known as DNP effect. The OMRI and PEDRI can observe DNP effect in vivo. Saturated the electron spin transition is necessary to obtain enhancement of tissue ^1^H signals. However, the oxygen in the sample can induce relaxation and interrupt the DNP process. Therefore, DNP MRI can reflect the oxygen concentration in the subject sample onto the image intensity. From several images observed with different EPR power, quantitative oxygen mapping can be obtained (Figure 7). Past studies reported in vivo DNP based oxymetric imaging were working at relatively low magnetic field. The 8.1 mT magnetic field for EPR excitation was immediately switched to the 15 mT magnetic field for MRI scan in the OMRI instrument. PEDRI instrument employed fixed 20.1 mT for both EPR and MRI. More details of DNP based oximetric imaging techniques have been described in other review paper [77].

## 5. MRI Based Oxygenation Imaging

BOLD [38,39,40] signals can detect T_2_* and tissue oxygen level dependent (TOLD) [79] signals can detect T_1_ contrast in MRI scans based on the variation of oxygen concentration in the subject now known as Oxygen enhanced MRI. Though Oxygen enhanced MRI is not able to quantify oxygen itself, the MR image intensity could be affected by oxygen concentration in the subject.

For BOLD MRI, several review papers were already published and described technical details with numerous applications in functional MRI [80,81,82]. Paramagnetic deoxyhemoglobin has a ferrous iron on the heme. The paramagnetic nature of deoxyhemoglobin gives inhomogeneity of magnetic field and makes proton T_2_* shorten. When deoxyhemoglobin in blood is oxygenated to diamagnetic oxyhemoglobin, the inhomogeneity of magnetic field would be diminished and the T_2_* getting longer. As a result, oxygenation in the blood vessels gives enhanced intensity in a T_2_* weighted MRI scan. BOLD effect has been utilized to investigate brain cortex functions due to that activation of brain functions could increase the oxygen transportation to the corresponding region in the brain [83,84]. Now the BOLD signal detection was applied not only for investigating brain function but also sensing functions of other tissues, such as lung, heart, kidney, tumor, and, etc. [85,86,87,88].

On the other hand, the TOLD signal is direct T_1_ shortening by the paramagnetic O_2_. As described above, molecular oxygen is a paramagnetic species with two unpaired electrons, which has T_1_-shortening effect, and can serve as a contrast media for a T_1_-weighted image. Increasing O_2_ concentration in the sample and/or subject, gives enhancement of T_1_-weighted signal in a T_1_-weighted MRI scan. Exposing a mouse bearing SCC tumor on the hind leg to the hyperbaric O_2_ atmosphere (2 atm) gives T_1_-weighted signal enhancement in the tumor tissue, however decreasing T_1_-weighted signal was observed in the normal muscle [79]. Figure 8 shows an example of TOLD MRI observed in tumor bearing mouse leg under hyperbaric O_2_ challenge. Mechanism of the negative TOLD signal in the normal tissue is still in progress.

Recently, comparison studies of BOLD and TOLD response in tumor tissues to an oxygen challenge have been continued by several groups [89,90,91]. For tumor tissue, it may be due to immature vasculature in tumor tissue; TOLD signal may be more responsible compared to BOLD signal. TOLD signal, which is variation of tissue T_1_-weighted signal, is not a quantitative value; however, the variation of R_1_ value, which is the reciprocal of T_1_, is quantitative [79,92].

## 6. Imaging Tissue Redox Status (Redox Imaging) Using Redox Sensitive Nitroxyl Contrast Agents

Nitroxyl radicals, which are also called nitroxides conventionally or called aminoxyl radicals formally on IUPAC, are relatively stable free radical species in solid form or when dissolved in solvent. Nitroxyl radical compounds have an un-paired electron on the molecule, with a π electron orbital formed on the nitrogen and oxygen. The nitroxyl radical compounds readily react with other free radical species. The nitroxyl radicals are mainly one-electron reduced to the corresponding hydroxyl amine form enzymatically in living organisms (Figure 9). Excessively generated ROS in cells/tissues can modify the in vivo reduction of nitroxyl radicals. Clinical and biological applications of nitroxyl radicals are well summarized in several review papers [93,94,95,96].

Redox imaging is a diagnostic imaging technique that use metabolically responsive T_1_ contrast agents to report on tissue redox status [78,98,99]. While Gd^3+^ based T_1_ contrast agents do not participate in redox reactions and hence can be used for perfusion studies only, nitroxyl radicals can act as T_1_ contrast agents and also participate in redox reactions to lose the T_1_ contrasting ability. Thus nitroxyl radicals, are used as a redox sensitive contrast agent for the redox imaging technique by monitoring the kinetics of the loss of paramagnetism while participating in intracellular redox reactions. Thus nitroxyl based redox imaging is a kind of dynamic imaging in which the time course of nitroxyl radical signal is observed by rapidly repeating the image acquisition. The redox imaging has initially developed in a field of in vivo EPR imaging [30,100]. However, the spatial resolution of EPR imaging is not high enough to distinguish particular organ/tissue due to relatively broad EPR line width of nitroxyl contrast agent, lack of anatomical information, and difficulty for selecting a particular slice as MRI and OMRI can do. The temporal resolution of EPR imaging was minutes based on CW modality. In addition, the EPR technique is currently limited to small animals only.

It has been known that nitroxyl radicals in an aqueous sample can be also detected by MRI through enhanced T_1_-weitghted contrast due to its proton T_1_ shortening effect. In fact, feasibility of nitroxyl radicals as T_1_ contrast agents in MRI has been examined in early 1980’s [101], that was before their use for EPR imaging. T_1_-weighted gradient echo MRI can provide an in vivo redox mapping based on the reduction of nitroxyl radical with fine spatial resolution and temporal resolution [102]. Using a combination of a nitroxyl contrast agent and MRI, spatially and temporally high-resolution redox mapping of particular slice of an animal can be achieved with excellent quality anatomical information. In addition, this MRI based redox mapping technique can be easily applied to larger animals and/or human patients.

Tumor tissue environment, i.e., low oxygen concentration, low pH, higher glutathione concentrations, are favorable conditions to reduce nitroxyl radical. Low oxygen environment can suppress reoxidation of hydroxylamine, which is one-electron reduced form of nitroxyl radical, back to nitroxyl radical, i.e., suppressing the left arrow on the base of triangle in Figure 9. Low pH environment may exaggerate one-electron oxidation of nitroxyl radical to oxoammonium cation, i.e., accelerating the up arrow on the left of triangle in Figure 9. High levels of glutathione can facilitate two-electron reduction of oxoammonium cation to hydroxylamine, i.e., accelerating the down arrow on the right of triangle in Figure 9. Therefore, apparent in vivo one-electron reduction of nitroxyl radical to hydroxylamine might be boosted in hypoxic tissue such as in tumors.

Mapping in vivo nitroxyl decay rate in SCC tumor loaded on a mouse thigh was measured using MRI [102]. Figure 10A shows the location of the axial slices of a mouse including the SCC tumor and the normal leg. T_1_-contrast was enhanced in both normal and tumor tissues after the administration of carbamoyl-PROXYL, and then gradually decreased (Figure 10B). The difference between tumor tissues and the normal tissues remained around the tumor tissues can be also seen clearly (Figure 10C,D).

Pharmacokinetics of three different nitroxyl contrast agents with different membrane permeability were tested [103]. Figure 11 shows pharmacokinetic (reduction) profiles of three nitroxyl contrast agents in SCC tumor, normal muscle, blood, and kidney observed by T_1_-weighted MRI experiment (circles), and total, i.e., both reduced and oxidized forms, contrast agent remaining in the tissues (diamonds). Reduction profile (T_1_-weighted MRI signal decay) of membrane impermeable contrast agent, carboxy-PROXYL, showed almost similar decay curve as the decay curve of total contrast agent. This result suggests that the MR sigal decay of membrane impermeable contrast agent is due to the clearance. The decays of the MR signal of membrane permeable nitroxyl contrast agents, TEMPOL and carbamoyl-PROXYL, are faster than that of total contrast agent, and this fact suggest that the decays of the MR signal of membrane permeable nitroxyl contrast agents reflect reduction of nitroxyl radical to corresponding hydroxylamine form. Table 1 shows decay rates of nytroxyl-induced T_1_-contrast in tumor and normal muscle. Membrane permeable nitroxyl radicals are reduced faster in tumor compared to normal tissue. A combination of a membrane permeable nitroxyl radical and a dynamic scanning T_1_-weighted MRI can give tissue redox information on the image.

Zhelev et al. [104] reported that normal tissues of tumor bearing mouse showed degradation of redox status. The tissues of healthy mouse showed rapid MRI signal loss of nitroxyl contrast agent, indicating a high reducing activity, however; that of tumor bearing mouse was slow. They also demonstrated that the re-oxidation of hydroxylamine to nitroxyl radical was faster in normal tissue of cancer bearing mouse compared with health mouse. The normal tissues of cancer bearing mouse was in oxidative circumstance, as a result the larger and sustained MRI signal could be observed in the normal tissues of cancer bearing mouse. They call this redox-imbalance. High cholesterol diet induced redox-imbalance in kidney was visualized using MRI and mito-TEMPO as a contrast [105]. EPR spectroscopic analysis in cells using several type of nitroxyl radical probes would be important to validate the phenomenon observed by MR redox imaging [106].

## 7. Nitroxyl Radical as Radioprotector (Contrast Agents Having a Medicinal Effect)

One of nitroxyl radicals, TEMPOL, was proposed as a normal tissue selective radioprotector. The nitroxyl radical form of TEMPOL can have the radioprotective effect, however, the hydroxylamine form does not have radioprotective effect directly [107]. On the other hand, an administration of TEMPOL-H to mice showed a similar radioprotective effect as TEMPOL [108]. This is because the TEMPOL-H is re-oxidized to TEMPOL in vivo. The pharmacokinetic curves of free radical form in blood, after the administration of TEMPOL-H and that of TEMPOL, are similar 15 min after the administration and later [108].

TEMPOL protected C3H mice against whole-body radiation-induced bone marrow failure [109]. Tumor growth curves generated after 10 and 33.3 Gy doses of radiation showed no difference in growth between the TEMPOL- and PBS-treated animals [110]. TEMPOL was evaluated for potential differential radiation protection of salivary glands and tumor using fractionated radiation [111]. Reduction in saliva production caused by five daily 6 Gy fractions to mouse head region was significantly protected by five daily treatments of TEMPOL. However, TEMPOL treatment had no effect on the radiation-induced inhibition of tumor regrowth. The hypothesis, how does the differential radioprotection between normal and tumor tissues occur, is understood as that TEMPOL is reduced or cleared from tumor tissue faster than normal tissues.

Contrast agent used in previous diagnosis imaging for planning radiotherapy may have a chance to be a normal tissue selective radio protector for the ensuing radiation therapy. This is one of application of theranostics, and possible applications were described below.

## 8. Applications of Redox Sensitive Nitroxyl Contrast Agents and Multimodal Contrast Agent

It has been already found that higher GSH content in tumor tissues accelerate reduction of nitroxyl radicals [112]. In addition, hypoxic environment in tumor tissues eliminate reoxidation of a hydroxylamine to the corresponding niroxyl radical, and consequently the apparent reduction rate markedly increased. The reduction rate of a nitroxyl radical increased as a function of tumor size [113]. This aspect of the use of nitroxyl contrast agents would be to exploit the markedly higher reduction rates in tumors compared to normal tissues in terms of diagnosis.

This different redox environment of nitroxyl radicals between normal and tumor tissues makes concentration of nitroxyl radicals between normal and tumor tissues different after the administration. Concentration of the free radical form in the tumor tissue rapidly decreased; however, concentration of the free radical form in the normal tissue was kept slightly higher due to re-oxidation of hydroxylamine form. Therefore, the nitroxyl contrast agents can be normal tissue selective radioprotector at the radiation therapy which may be carried out after the diagnosis. Recently it was found that 100% oxygen breathing can accelerate reduction rate of membrane permeable nitroxides in tumor tisse, and make large difference of nitroxyl radical concentration between normal and tumor tissues [91].

Another possibility of nitroxyl contrast agents is in brain molecular imaging. There are several nitroxyl radicals (Figure 12), which can go through the blood-brain-barrier [114,115,116]. Nitroxyl radical induced T_1_ contrast at a head part of mice show different distributions of nitroxyl contrast agents in brain depending on its membrane permeability (Figure 13). Highly permeable TEMPOL and methoxycarbonyl-PROXYL showed high T_1_ contrast induction in whole brain area. Effect of irradiating X-ray or carbon-ion-beam to mouse brain on the brain redox status was investigated [117]. Different redox responses in the brain were induced by X-ray and carbon-ion-beam irradiation. Functional brain imaging experiments also have been performed using an EPR imaging scanner [118,119,120].

An anti-cancer drug, lomustine, labeled by nitroxyl radical was synthesized as a BBB permeable anti-cancer contrast agent, and distribution of this new drug in mouse brain was visualized by MRI [121]. This nitroxyl radical labeled drug, spin-labeled nitrosourea (SLENU), has an anticancer effect. The approach is applicable not only for non-invasively visualizing distribution of the drugs in the target tissue/organ, but also for imaging redox status in the target and surrounding tissues/organs. Another progress of development of nitroxyl radical labeled theranostic compounds has been reported, such as nitroxyl labeled ibuprofen, ketoprofen [122] and nitroxyl labeled theophylline [123]. Contras agent having medicinal benefits can be a useful tool in theranostics approach, i.e., efficacy as a drug, visualizing drug distribution, and may have an additional chance to make a diagnosis based on the tissue redox status. In addition, anticancer effects have been established for some of nitroxyl radical compounds itself [124].

Assessing the tumor microenvironment, such as pH, pO_2_, redox status, and concentrations of phosphate and glutathione, can be even performed on single instrument, in vivo EPR, specific molecular probe for detecting each effect was required [125]. Future development of a suitable specific contrast agent for each imaging modality will be a key for the success of theranostics.

Actually, a nitoxyl radical is a kind of multimodal contrast agent, which can be detected by either EPR or MRI as described above. A polymer loaded by both dense nitroxyl radicals for MRI and a near-infrared fluorophore for optical imaging were synthesized, and in vivo dynamics and imaging ability was demonstrated [126]. Tobacco mosaic virus as a core modified by both nitroxyl radicals outside and by fluorophores inside was reported as a superoxide sensor working eather on MRI and EPR [127]. Gold nanorod loading nitroxyl radicals for dual imaging of X-ray CT and MRI was developed for redeeming insufficient contrast in soft tissue by X-ray and contrast stability of MRI [128].

Nitroxyl radical labeled polymer type contrast agent for MRI called organic-radical contrast agents (ORCAs), which is structured by dendrimer core and lapping polyethylene glycol chains, has been proposed by Rajca et al. [129]. For next development design of ORCA, nitroxyl radical, fluorophore, prodrug was loaded on the lapping polymer and the polymers were conjugated on core [130,131,132]. In other words, this assemble type contrast agent is working as drug carrier simultaneously, and then, monitoring of the loaded drug from the assembled compound were reported [132]. Nitroxyl radical labeled contrast agents as protein [133] and amino-acid type [134] for achieving simultaneous drug delivery and tumor imaging has been reported. A nitroxyl radical labeled proteolysis probe for OMRI has been also proposed [135]. The nitroxyl radical connected on big protein molecule can not give OMRI enhancement, but free nitroxyl molecules released can give OMRI enhancement. Such nitroxyl labeled polymer and/or proteins are expected as a theranostic contrast agent.

## 9. Metabolic Imaging and Multimodal Comparison

In aerobic energy production process, pyruvate generated from glucose is converted to acetyl-CoA, and then acetyl-CoA is metabolized to CO_2_ and H atoms in the citric acid cycle. Therefore, O_2_ is required to maintain reactions in the citric acid cycle. However, in hypoxic condition, pyruvate is metabolized to lactate instead of getting into the citric acid cycle. The glucose breakdown to lactate through pyruvate is so called hypoxic glycolysis. During aerobic energy production process, also called oxidative phosphorylation, working with the citric acid cycle, 38 ATP can be generated with consuming 1 glucose, however; through hypoxic glycolysis, only 2 ATP can be generated from 1 glucose. Therefore, hypoxic tumor cells need more glucose to make energy to survive.

Excess glucose uptake of tumor/cancer tissues has been visualized using ^18^F-DG PET [28]. On the other hand, ^13^C-DNP MRI can visualize glycometabolism of tumor/cancer tissues [23]. Hyperpolarized ^13^C-labeled pyruvate was metabolized to lactate by the dominant glycolytic system in the hypoxic tumor tissue. Lactate or other metabolite of pyruvate can separately visualized by chemical shift of resonance peaks. The pyruvate showed the single line at 173 ppm, and the lactate showed the line at 185 ppm (right panel of Figure 14B).

It can be expected that the hypoxic region and glycolytic region of tumor tissue should be overlapped. Experimentally it was confirmed that hypoxic region and glycolytic region was actually overlapped by co-registering the pulse EPR pO_2_ mapping and ^13^C-labeled pyruvate metabolic image [136]. In this study, local metabolic changes in hypoxic area induced by radiation were investigated by co-registration analysis of EPR oxygen mapping and metabolic MRI by hyperpolarized ^13^C-labeled pyruvate (Figure 14). Hypoxic region showed elevated hypoxic metabolism, i.e., more lactate generation.

In vivo mechanisms of the anti-angiogenic drug sunitinib to transiently improve oxygenation by vascular renormalization was investigated by multimodal imaging [137]. Administration of sunitinib, suppressed tumor growth, suppressed blood volume in tumor tissue measured by using MRI with blood pooling T_2_ contrast agent USPIO, increased Gd-DTPA uptake into tumor tissue observed by T_1_-weighed MRI, and improved tumor oxygenation visualized by EPR oxygen mapping. It was also observed by the metabolic ^13^C MRI that the sunitinib suppressed glycolytic metabolism of pyruvate to lactate (Figure 15). In addition, sunitinib-induced oxidative shift, i.e., make reduction of nitroxyl radical slower, in tumor tissue was observed by MR redox imaging. The normalization of tumor blood flow by sunitinib made pathophysiological function of tumor tissue shifted to normal side.

Multimodal imaging analysis can provide a correlation among a treatment given and relation of resulted biological effects clear. Information obtained from single imaging modality is limited. Most of imaging modality can detect only specific information dedicated to the instrument used. However, the pathological condition being monitored is probably a complexing condition of several biological effects. The fusion/co-registration of three digital imaging techniques, MR redox imaging, EPR oxygen mapping, and hyperpolarized ^13^C MRI techniques in the magnetic resonance field, can widely contribute to the theranostics.

## 10. Future Directions

Translational or preclinical multimodal imaging techniques combining PET, EPRI, and/or MRI, would be useful to monitor abnormal tissue/cell microenvironments such as tumor hypoxia, oxygenation ability, redox status, and/or glycolmetabolism. Not only accurate co-registration technique among several different modalities using fiducial markers but also combined system of different instruments will be a valid and feasible way. Combination system of PET and MRI is already commercialized for clinically. The PET/MRI combination may have grate advantage to detect multiple biological functions, such as disordered redox status, hypoxia, glycolmetabolism, , etc., with clear anatomical mapping.

Another expected development is probably a multimodal contrast agent, which can be worked as a drug itself or a drug carrier, and simultaneously detect one or more biological events after a single dose. A hybrid contrast agents, which can detected by two or more instrumental combination will be also useful tool for achieving theranostics.

Development of medical imaging techniques is an optimal combination of multiple fields. Instrumental development requires physical, physical-chemical, and medical knowledge and technologies. Development of a contrast agent requires physical-chemical, bio-chemical, pharmacological, and medical knowledges, technologies, and the pharmaceutical approving procedures. Development of experimental models requires medical, veterinary, and pharmacological approaches. The greatest need at the present stage of most medical imaging techniques for future theranostic use may be development of a highly specific contrast agent for visualizing target biological phenomenon efficiently. Multimodal analysis with multiple perspectives will be also important for finding a solution from a complex system.

## 11. Conclusions

Imaging technique with multimodarity of detection is required to visualize both biological effects making the cause of the disorders and the response of the cells/tissue systems with accurate anatomical information, simultaneously or sequencialy. PET is sensitive imaging method probing by contrasting positron-emission radioisotopes. PET can detect not only for tissue glycodynamics or hypoxia but also have possibility to sense most of everything adapting on positron-emission nuclei labeled probes. EPRI is the only imaging modality which enable us a quantitative in vivo oxygen mapping in preclinical animal models. Magnetic resonance imaging is kind of multimodal tool to detect multiple biological functions, such as disordered redox status, hypoxia, oxygenation, glycolmetabolism, , etc. Multimodal combinations of imaging techniques, such as PET, EPRI, and/or MRI, would be a useful tool to make an analytic and accurate diagnosis for future theranostics.

## Figures and Tables

**Figure 1 molecules-26-01614-f001:**
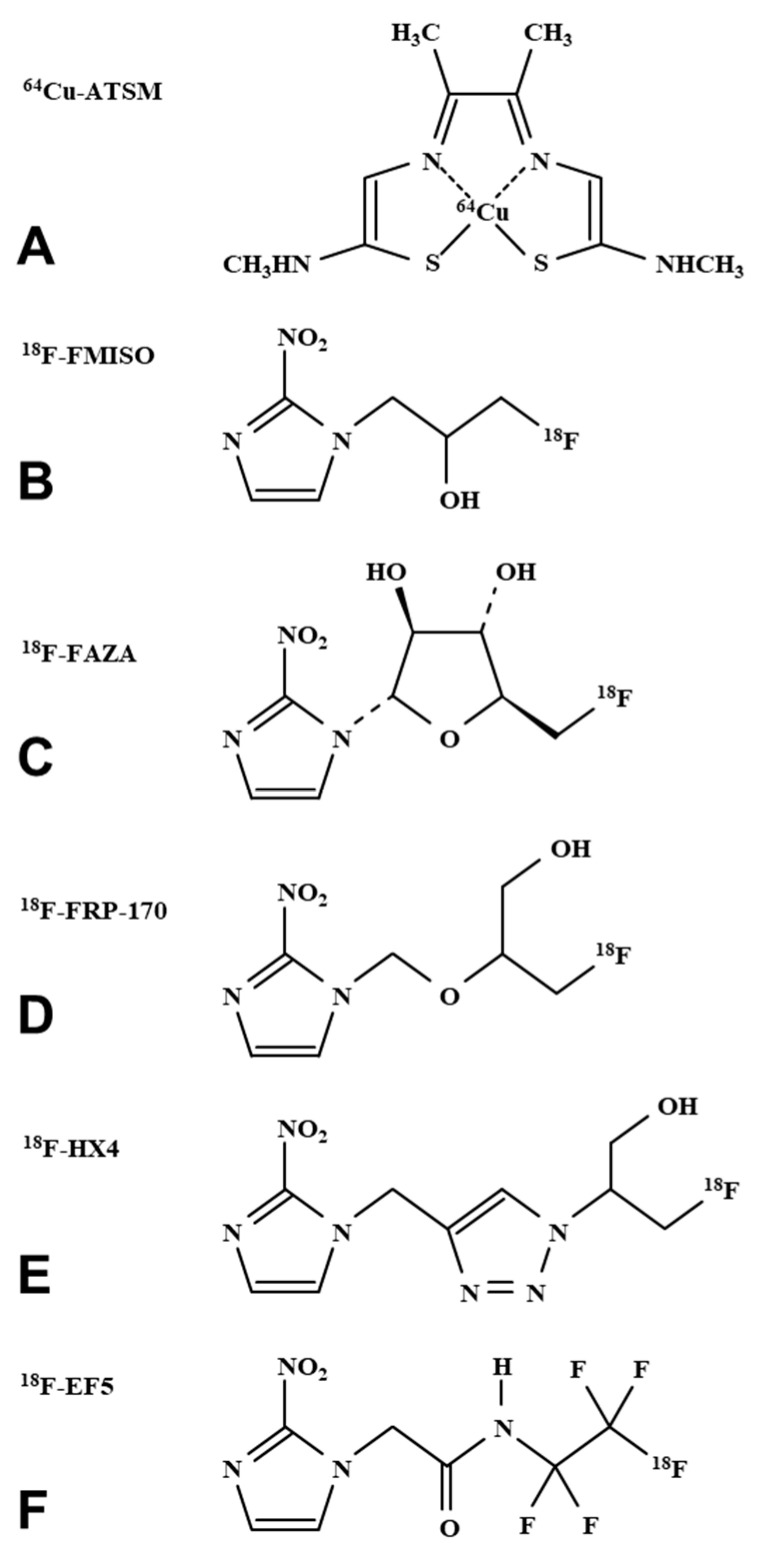
Chemical structures of PET probes seeking tissue hypoxia. (**A**) ^64^Cu-ATSM has a chelated radioactive ^64^Cu^2+^ on the ATSM, which is a1 copper chelator. (**B**) ^18^F-FMISO, (**C**) ^18^F-FAZA, (**D**) ^18^F-FRP-170, (**E**) ^18^F-HX4, and (**F**) ^18^F-EF5 were all labeled by a radioactive ^18^F and have a nitroimidazole ring on the molecule.

**Figure 2 molecules-26-01614-f002:**
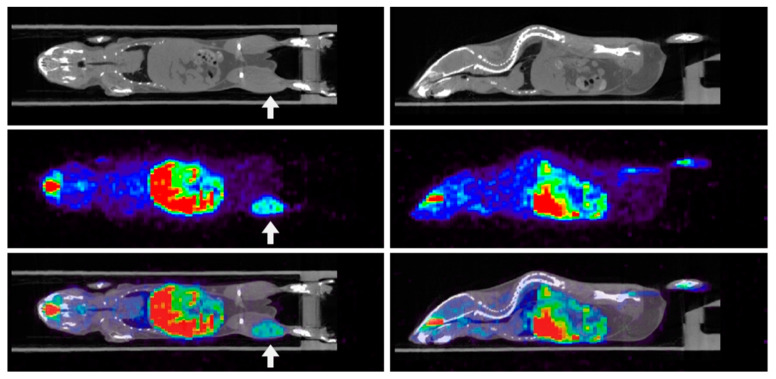
Whole body distribution of ^64^Cu-ATSM. Coronal (left 3 panels) and sagittal (right 3 panels) whole body scans of a SCCVII tumor-bearing mouse. Top, center, and bottom images show X-ray CT image, 64Cu-ATSM PET image; fused ^64^Cu-ATSM/CT image, respectively. Uptake of ^64^Cu-ATSM was obtained in the tumor-bearing leg (upper arrow in coronal images), abdomen (liver and duodenum intestine), and head region of the animals. The figures were partly modified from our previous reports [52].

**Figure 3 molecules-26-01614-f003:**
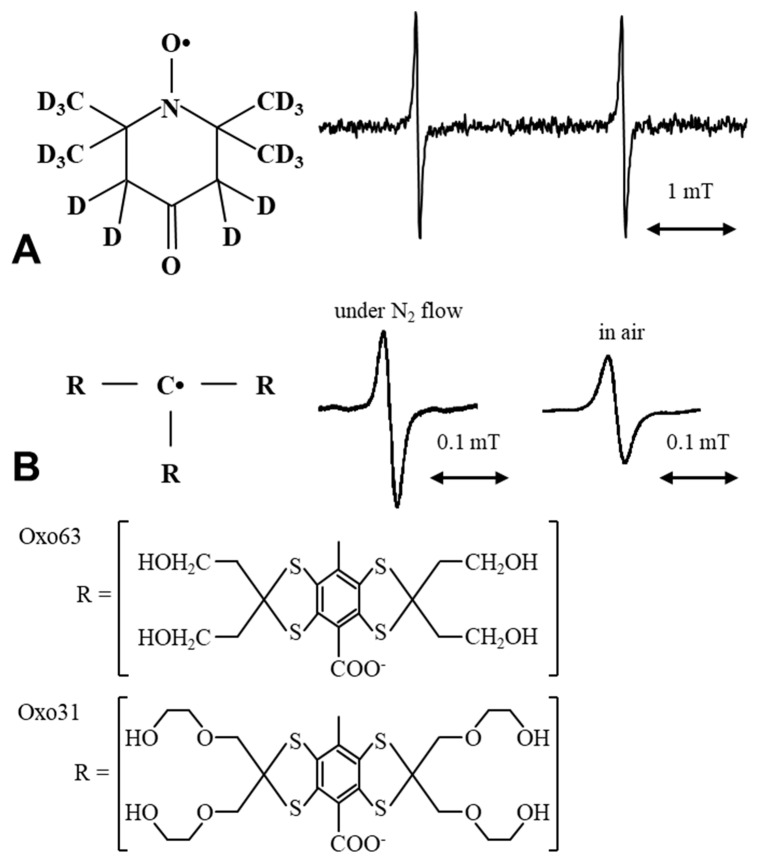
Chemical structures of oxygen probes using EPR based oxygen mapping. (**A**) [^15^N]PDT (4-oxo-2,2,6,6-tetramethyl [1-^15^N]piperidine-D_16_-1-oxyl) is an ^15^N labeled and deuterated analog of a nitroxyl radical called oxo-TEMPO or called TEMPONE. X-band EPR spectra of [^15^N]PDT showed doublet resonance lines. (**B**) Triarylmethy radical has narrow single line EPR spectrum. Inserted spectra are of Oxo63 measured under N_2_ gas flow or under air atmosphere. Oxo31 has narrower EPR linewidth compared to that of Oxo63.

**Figure 4 molecules-26-01614-f004:**
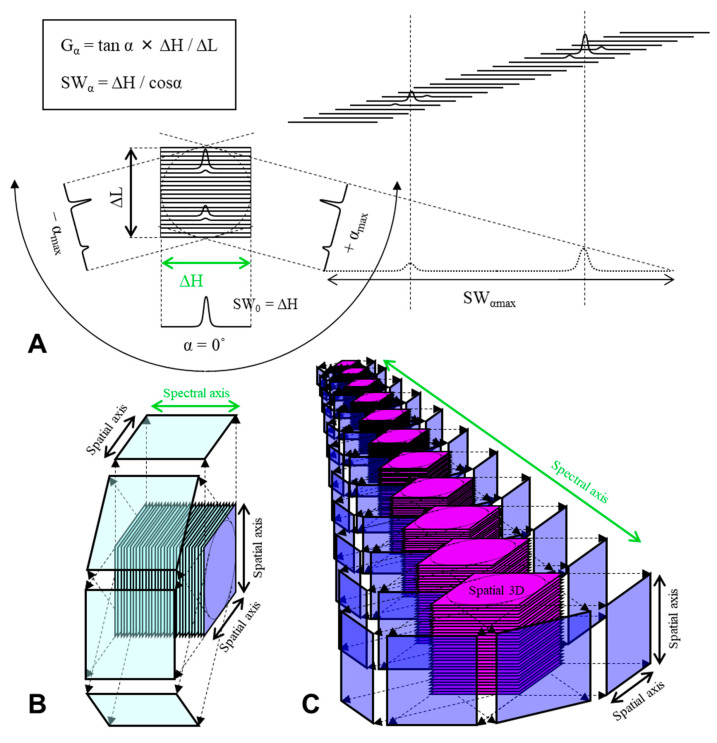
Projection reconstruction of 2D, 3D, and 4D spectral-spatial image. (**A**) Theoretical scheme of spectral-spatial 2D imaging in frequency domain. The projection data are collected using unidirection but incrementing magnitude of field gradient. G, magnitude of field gradient (Gauss/cm); ΔH and ΔL, spectral and spatial window width of the pseudo spectral-spatial matrix; SW, sweep width; α, viewing angle on the pseudo spectral-spatial matrix; H, magnetic field; L, spatial length. SW is varied depending on the α. Rotating field gradient direction achieves (**B**) 3D spectral-spatial (1D spectral and 2D spatial) or (**C**) 4D spectral-spatial (1D spectral and 3D spatial) imaging. The figures were partly modified from our previous reports [74].

**Figure 5 molecules-26-01614-f005:**
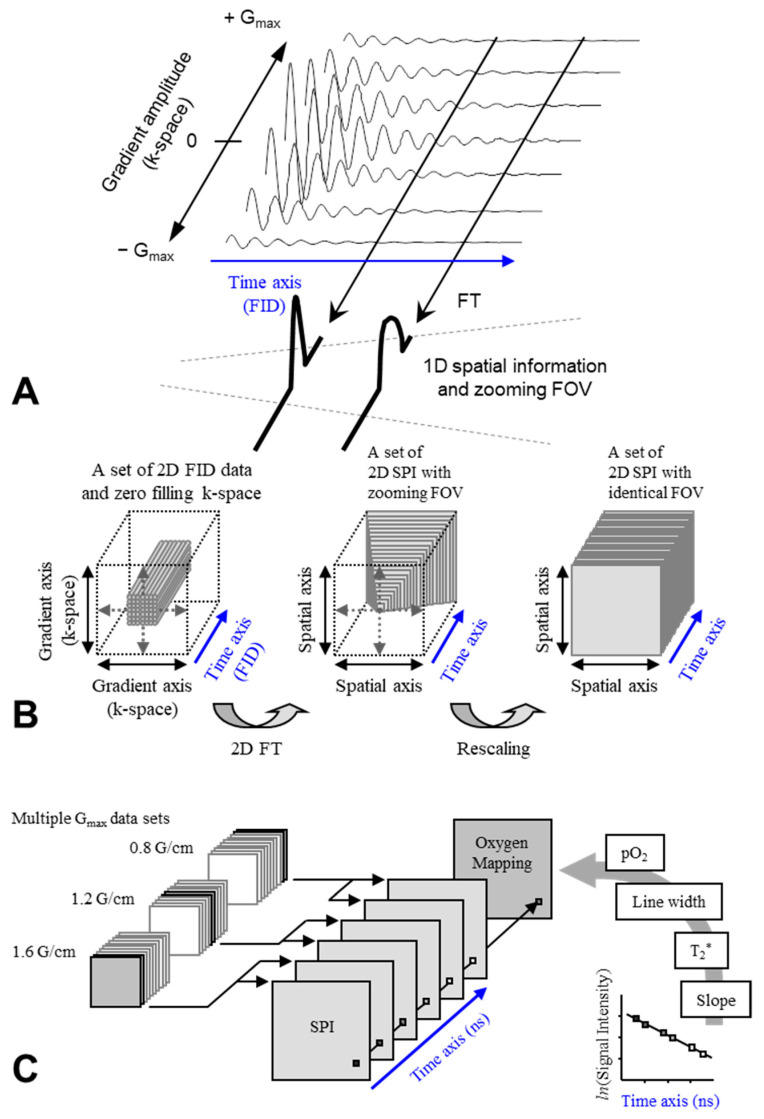
Spectral-spatial imaging in time domain. (**A**) A schematic drawing of theory of SPI basis 2D spectral-spatial imaging. Spectra (FIDs) are collected using incrementing but unidirectional field gradient strengths and a constant time window (sweep width in frequency domain). Fourier transformation along G axis gives a spatial profile. The FOV of the spatial profile is varied depending on the time. With combinations of two or three orthogonal field gradient set, 3D or 4D imaging is available. (**B**) A schematic drawing of data manipulation of 3D spectral-spatial imaging. Left: A set of FIDs measured under a 2D field gradient was placed on a 2D k-space and then the matrix was zero-filled (2n × 2n) for FT. Center: A set of 2D SPI, but delayed time points represent larger durations of the phase-encoding gradients and lead to lower Nyquist bandwidths corresponding to smaller FOVs (i.e., “zoomed-in” images). Gray slices indicate regions of identical FOVs for each time point. Right: All SPIs were rescaled to an identical FOV, and a 3D matrix (2D spatial and 1D time domain) was obtained. (**C**) Estimation of pixel-wise pO_2_ from SPI data sets. An SPI data set was reassembled from several SPI data sets obtained by using multiple Gmax settings, to obtain a certain image resolution along the time axis. Pixels of reconstructed FID are replotted semilogarithmically, and the slope of the semilogarithmical plot of the FID gives T_2_*. The EPR linewidth, and then pO_2_ can be calculated from the T_2_*. Finally, pO_2_ values are rearranged onto a matrix (oxygen mapping). The figures were partly modified from our previous reports [74].

**Figure 6 molecules-26-01614-f006:**
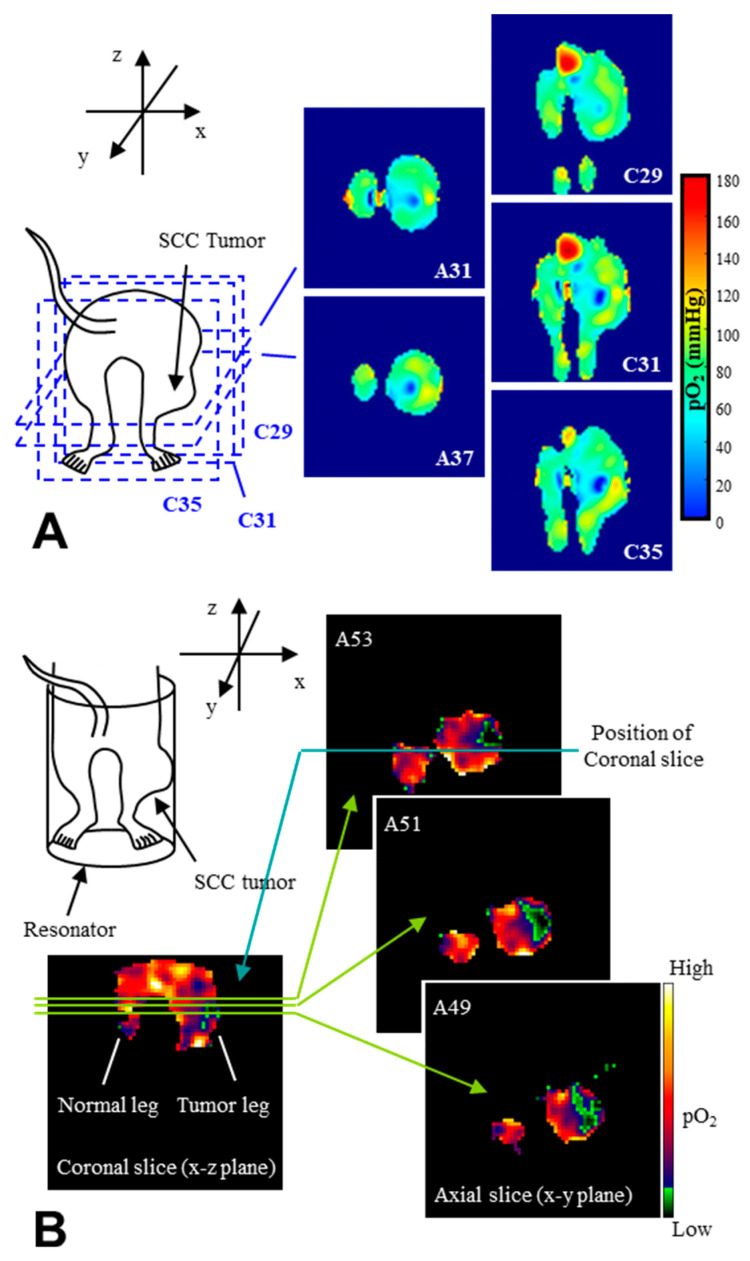
Examples of EPR basis in vivo 3D oxygen mappings. (**A**) 3D oxygen mapping observed by SPI based 4D EPR spectral-spatial imaging. The SPI data sets were obtained with three different Gmax settings (1.5, 1.2, and 0.8 Gauss/cm) and 21^3^ k-space samples were acquired in 18 min. The FOV was encoded in 21 gradient steps corresponding to a slice thickness of 2.2 mm. A 3D image was reconstructed on 64^3^ matrix, giving a voxel resolution of 0.7 mm^3^. The numbers on the image refer to the slice number. (**B**) In vivo power saturation 3D oxygen mapping of the tumor-bearing mouse. Power saturation image obtained by two different radiofrequency power levels (0.25 and 130 mW) clearly depicts the hypoxic foci of the SCC tumor on the hind leg of a mouse. Both methods using a triarylmethy radical, Oxo63, as the oxygen probe. The figures were partly modified from our previous reports [63,67].

**Figure 7 molecules-26-01614-f007:**
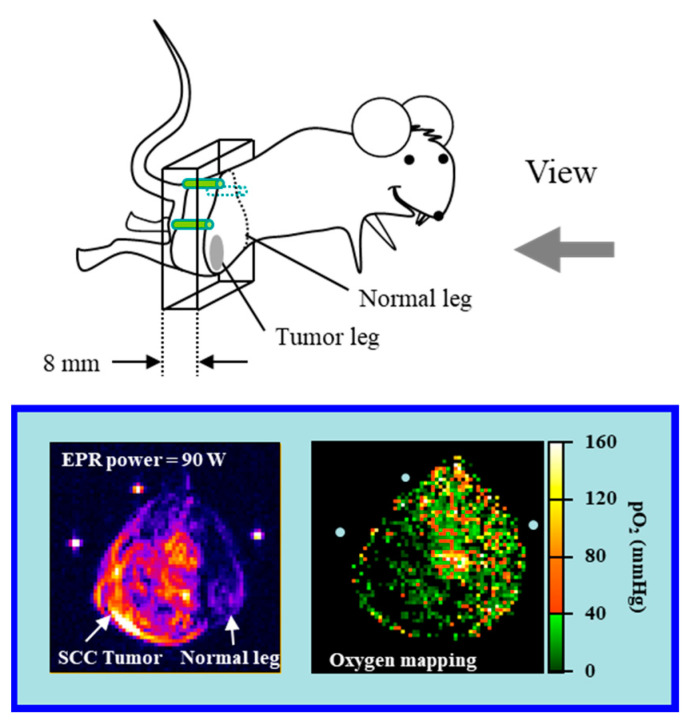
An examples of DNP basis in vivo oxygen mapping. Top: The direction of the slice view of OMRI image with respect to the subjected mouse. Bottom left: Spin-density image (raw OMRI image) shows the differential accumulation of the paramagnetic oxygen probe, Oxo63. Bottom right: Oxygen mapping of an axial slice of the tumor-bearing and normal legs of a mouse computed from OMRI images taken at two different EPR power levels. The figures were partly modified from our previous reports [78].

**Figure 8 molecules-26-01614-f008:**
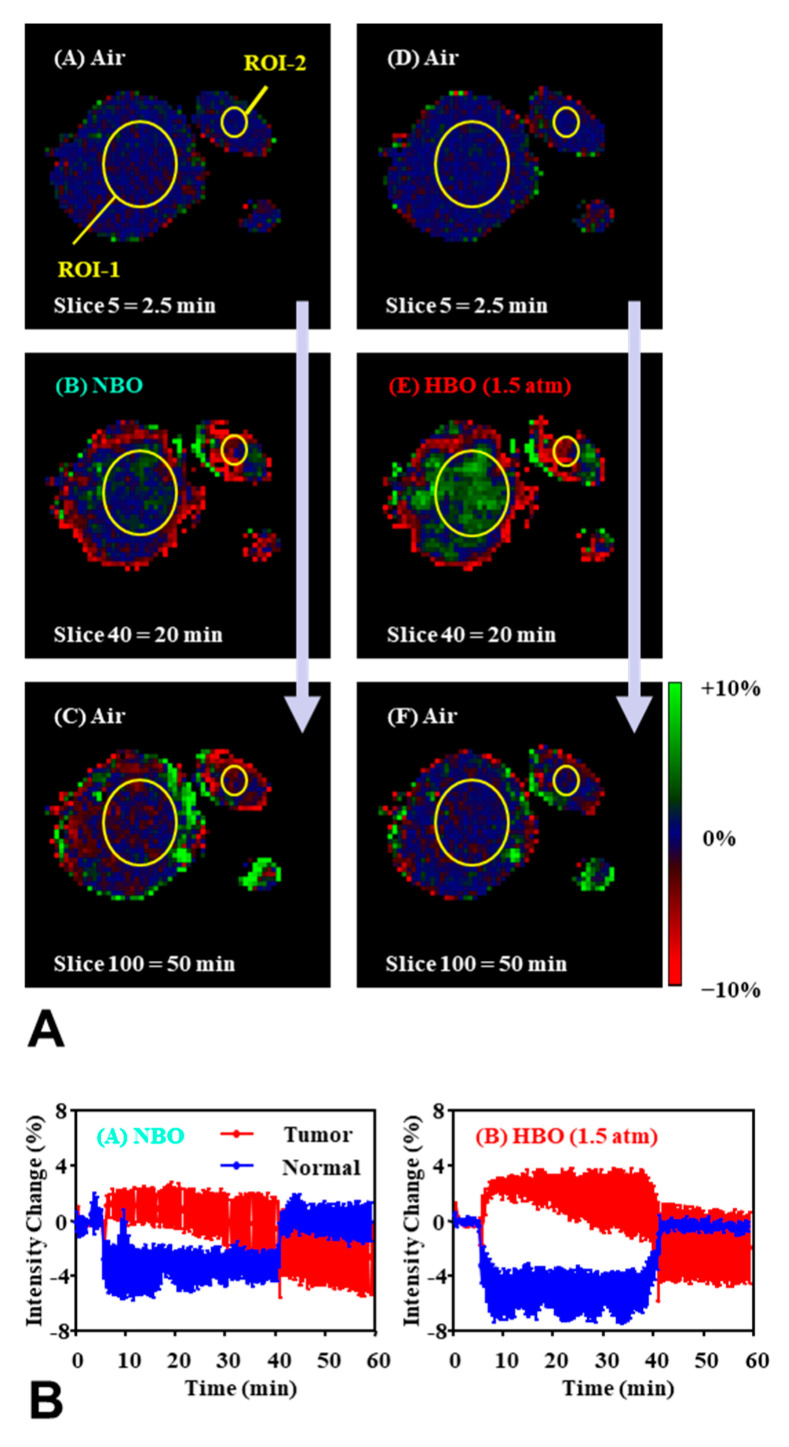
An example of TOLD MRI. (**A**) Comparison of T_1_ weighted MR images of SCC tumor bearing and normal leg of an identical mouse under (left column) normobaric and (right column) hyperbaric oxygen challenges. A mouse legs was scanned by GEFI sequence, and axial 2 mm slice reconstructed on 64 × 64 matrix was obtained every 30 sec for 60 min. The gas condition was switched from air to normobaric or hyperbaric oxygen 5 min after starting scan, kept for 35 min, and then switched back to air. The normobaric and hyperbaric experiments were sequenced with 20 min gap. Percent-changes of T_1_-weighted image intensities from baseline images were observed. No signals are observed under air flow (top panels). Positive and negative signals are observed in the center and periphery of tumor leg, respectively, under the tasks (center panels). Signals observed in center of tumor dropped down to base line level when breathing gas was switched back to air (bottom panels). (**B**) Time course of T_1_-weighted signal changes in tumor (ROI-1) and normal (ROI-2) legs under normobaric (left) and hyperbaric (right) challenges. The values indicated the average ± SD of results of 4 mice. The figures were partly modified from our previous reports [79].

**Figure 9 molecules-26-01614-f009:**
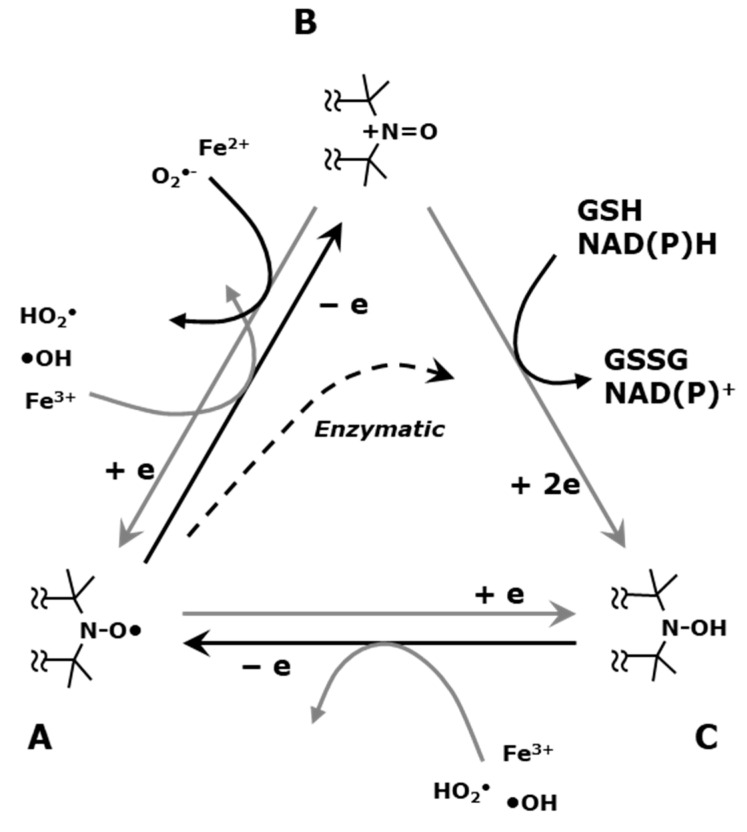
Redox transformations of nitroxyl radicals in (**A**) the free radical state, (**B**) the oxoammonium cation, and (**C**) the hydroxylamine. The nitroxyl radical could be one-electron oxidized to become the corresponding oxoammonium cation form. The oxoammonium cation could readily be back to nitroxyl radical by one-electron reduction, or two-electron reduced to hydroxylamine form by receiving a hydrogen atom from hydrogen donor, such as NAD(P)H and/or GSH. The hydroxylamine could be one-electron oxidized to be nitroxyl radical form. The figures were partly modified from our previous reports [97].

**Figure 10 molecules-26-01614-f010:**
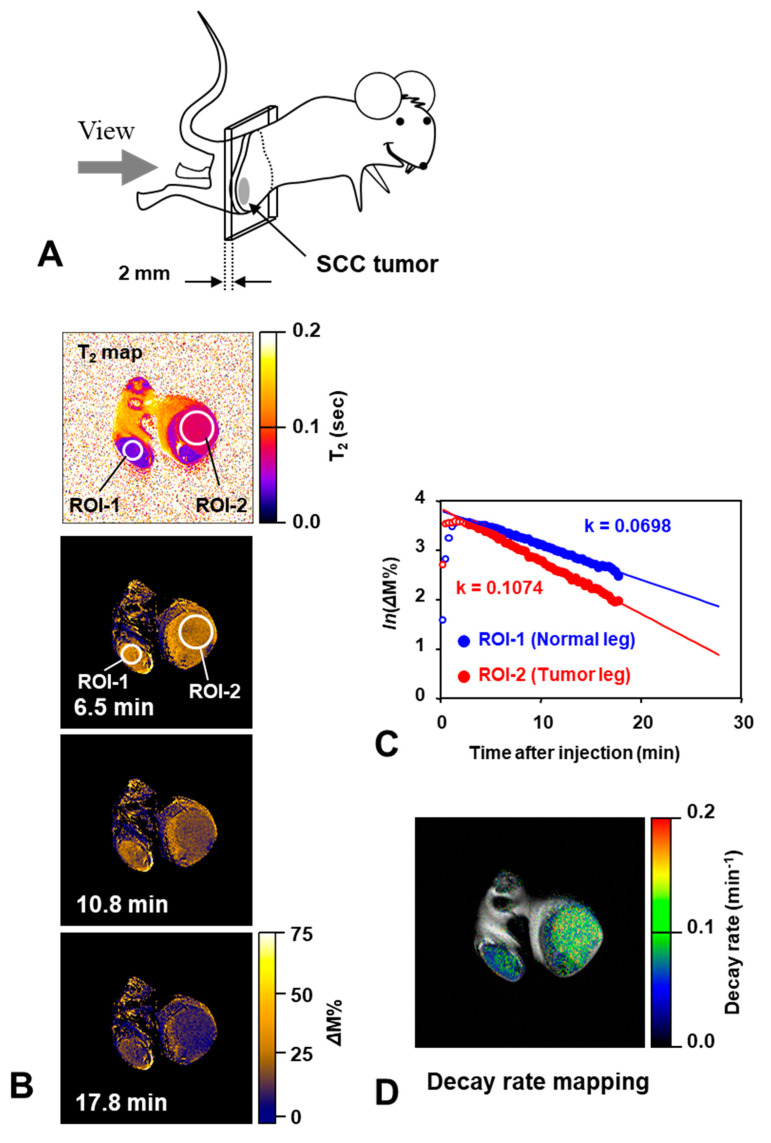
An example of MR-based redox imaging. (**A**) Direction of the slice view of MRI with respect to the subjected mouse. (**B**) Time course of ΔM% signal of T_1_-weighted MRI and a scout T_2_-mapping for ROI selection. Time after injection was indicated in each image. ROI-1 for normal leg and ROI-2 for tumor leg were estimated based on a previously obtained T_2_-mapping. Field of view was 3.2 × 3.2 cm. (**C**) Time course of average ΔM% signal in the ROI-1 and ROI-2. Logarithmic values of ΔM% signal in the ROIs are plotted with time. Decay rate constants were obtained from the slope of linear decay after peak. (**D**) Decay rate map overlapped on the corresponding multi-slice-multi-echo image can show a distribution of decay rates with clear anatomic information. The figure was partly modified from our previous report [102].

**Figure 11 molecules-26-01614-f011:**
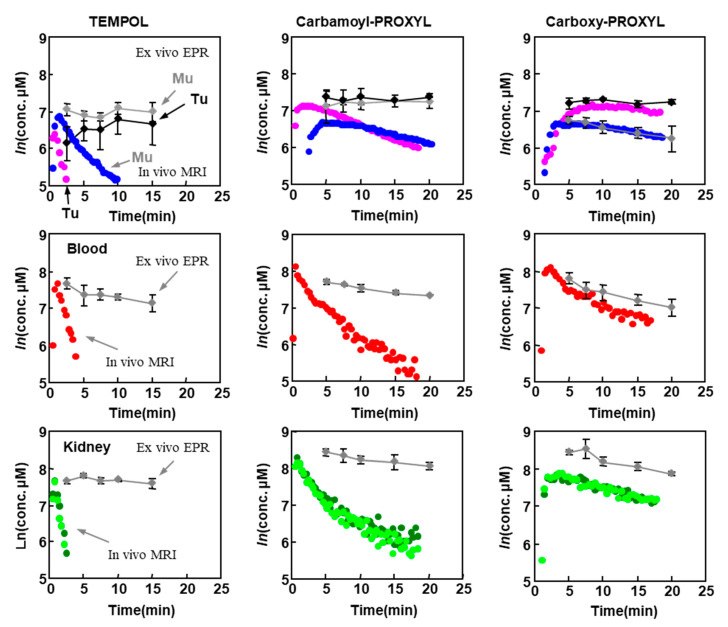
Comparison of pharmacokinetic profiles of three nitroxyl contrast agents by T_1_-weighted MRI. The pharmacokinetic profiles of oxidized form and total (nitroxyl radical form + hydroxylamine form) TEMPOL (left), carbamoyl-PROXYL (center), and carboxy-PROXYL (right). The time course of nitroxyl radical form in normal tissue (blue circle), tumor tissue (purple circle), blood (red circle), and kidney (left kidney, dark green circle; right kidney, light green circle) were obtained by T_1_-weighted MRI. The concentrations of total nitroxyl contrast agent (nitroxyl radical + hydroxylamine) measured by X-band EPR spectroscopy in the corresponding tissues are indicated by gray diamond or except black diamond for tumor tissue. The figure was partly modified from our previous report [103].

**Figure 12 molecules-26-01614-f012:**
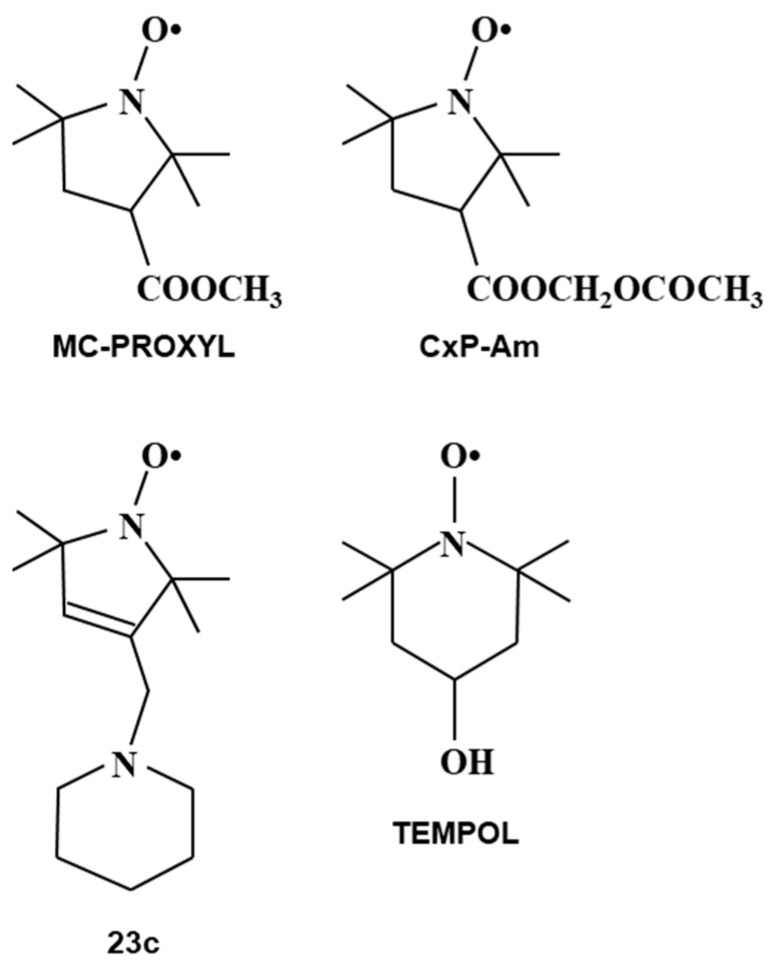
Chemical structures of membrane permeable nitroxyl radical contrast agents.

**Figure 13 molecules-26-01614-f013:**
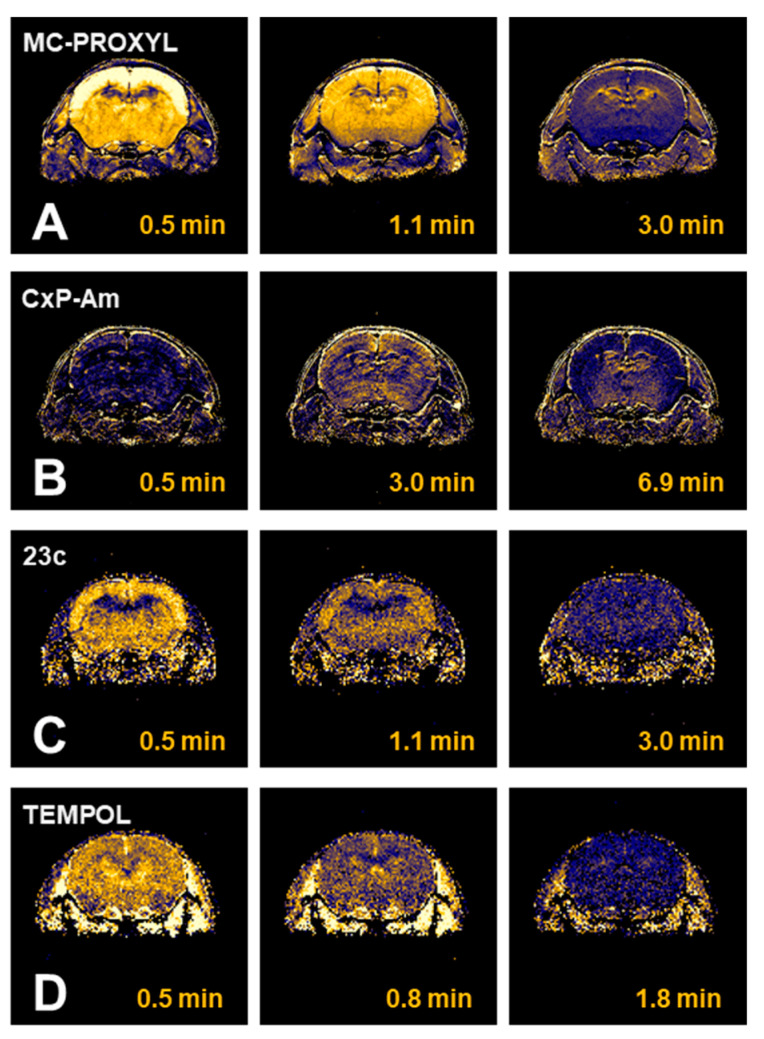
Distributions of nitroxyl contrast agents in mouse brain. T1-weighted signal enhancement images of mouse head were obtained after *i.v.* injection of (**A**) MC-PROXYL, (**B**) CxP-Am, (**C**) 23c, and (**D**) TEMPOL. Horizontal row showed the time course of the T_1_-weighted signal enhancement. The figures were partly modified from our previous reports [116].

**Figure 14 molecules-26-01614-f014:**
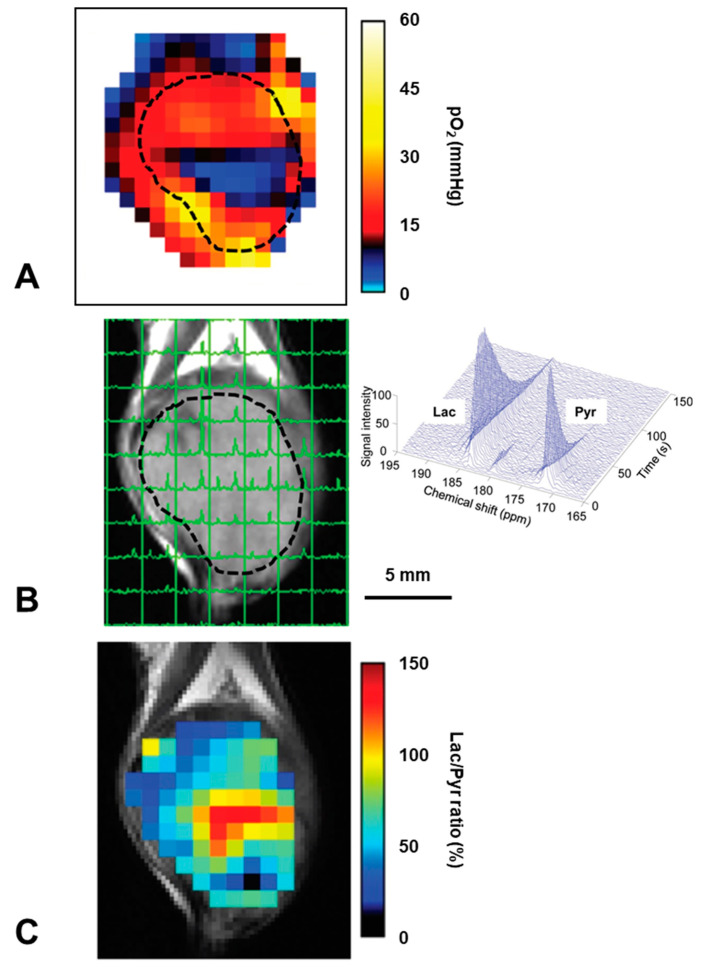
Comparison of the hypoxic region and glycometabolic shift in a SCCVII tumor on a mouse leg. (**A**) The pO_2_ map obtaine by pulsed EPR oxygen imaging shows a hypoxic core (blue) in the central part of the tumor. (**B**) Chemical shift imaging obtained by ^13^C-DNP MRI 12 sec after hyperpolarized ^13^C-labeled pyruvate injection is overlaid on the T_2_-weighted image. Each signal peak (green line) corresponds to pyruvate (right) and lactate (left), respectively. (**C**) The lactate/pyruvate ratio map shows higher lactate/pyruvate ratio area dominantly exists in the hypoxic region on EPRI. The figure was partly modified from our previous report [136].

**Figure 15 molecules-26-01614-f015:**
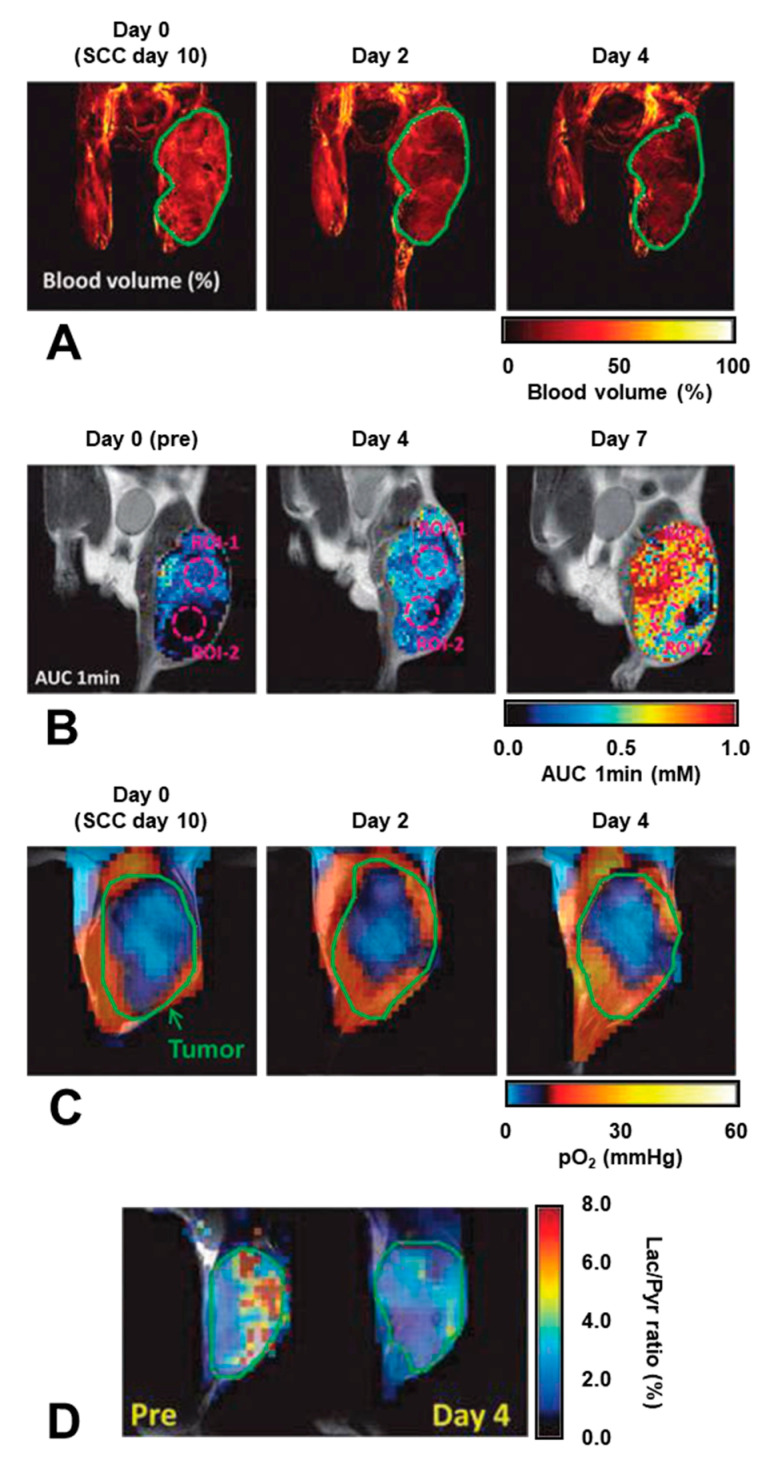
Effect of anti-angiogenic drug sunitinib treatment on tumor blood volume, Gd-DTPA uptake, pO_2_, and glycometabolism. (**A**) Blood volume images in SCCVII tumor on a mouse leg observed by MRI with blood pooling T_2_ contrast agent USPIO before and after 2 and 4 days of daily sunitinib treatment. (**B**) Area under the curve (AUC) images of the first minute after Gd-DTPA injection. (**C**) EPR oxygen images obtained before and after 2 and 4 days of daily sunitinib treatment. (**D**) The ^13^C labeled lactate/pyruvate ratio images calculated from the ^13^C chemical shift images. The figure was partly modified from our previous report [137].

**Table 1 molecules-26-01614-t001:** Comparison of In Vivo Decay Rates of Nitroxyl-Induced T_1_-Weighted MRI Intensity in Normal and Tumor Tissues.

Tissues	TEMPOL Decay Rate (min^−1^)	Carbamoyl-PROXYL Decay Rate (min^−1^)	Carboxy-PROXYL Decay Rate (min^−1^)
Normal muscle	0.319 ± 0.025	0.056 ± 0.013	0.029 ± 0.014
Tumor tissue	1.095 ± 0.203 **	0.107 ± 0.20 *	0.020 ± 0.014

** and * indicates significant difference between normal and tumor tissue as *p* < 0.01 and *p* < 0.05, respectively.

## Data Availability

Not applicable.

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
