# Peer review of "Multimodal Functional Imaging for Cancer/Tumor Microenvironments Based on MRI, EPRI, and PET"

_molecules, 2021, doi:10.3390/molecules26061614_

Round 1

Reviewer 1 Report

The review “ Theranostic Imaging for Cancer/Tumor based on Magnetic Resonance Imaging” by Ken-ichiro Matsumoto, James B. Mitchel and Murali C. Krishna aims to describe the molecular imaging of tumor “microenvironement” by several techniques: MRI, PET, EPRS ….. The subject is of importance as it merges the research, production and therapeutic (medical) fields. Without the understanding of the “responses” it would be difficult to have new probes, new equipment and improved therapies. The discussion revolves basically around the varying (or low) pO2 environment in the tumor surrounding and to what extent it can be “visualized” by the available techniques, using contrast agents, etc. In that sense the understanding is that Theranostic combines the therapy, follow up of the therapy, adjusting the therapy - here talking even personalized - with the help of the “visualization/imaging”. In the abstract the authors put the focus on Radiation therapy and the fact that “Tumor hypoxia could be sensed by PET, EPR 17oxygen mapping, and in vivo DNP MRI” . In the 1. Introduction the four paragraphs dealing with radiation therapy , Tumor microenvironment, Modern medical imaging techniques X-ray CT, PET, and MRI and again Hypoxia and/or redox are finished with the aim of the review P2L65.

If the focus is on multimodal imaging techniques then the title should be rewritten (not just MRI) . Actually, it would be even more suitable if the authors point out which multimodal are considered in the review. MRI/PET, MRI/EPR, MRI/CT etc. A paragraph of radiation therapy (or other therapy)  “effects” monitored by the mentioned  techniques would be welcomed. Not just pure detection. 

The organization of the review is a bit point by point. Actually 2. Development of Modern Medical Imaging Techniques is just a listing of the techniques. Has nothing to do with in-depth analysis of the techniques development. The title of Point 2 should revised ( may include the multi modal suggestion) or point 2  rewritten. 

 Then 3 to 5 are chapters that deal with PET hypoxia, EPR oxygen map, MRI Oxygenation. Those are easy to understand  but  are more self oriented.

6. Nitroxyl Radical is just an insert of a mechanism and is followed by 7. Redox Imaging . 

The real emphasis of the review is in p.8 -10, However those points do not include a suitable amount of references to cover in detail the subject(s) – references are needed, actually a lot of recent data is available and should be mentionned. The suggestion to the authors is to focus on recent references, they do not really need to go back to the 80s, probably just focus on the last 20 years were the IT/computer revolution went with the rapid progress of the "techniques".

Conclusions:   Those are extremely general. The authors may consider to  include future directions ?

In  the present state the manuscript is quite fragmented, describes mostly techniques,   needs the  addition of recent references and thus to point/emphasis the focus on the Theranostic- multimodal imaging aim of the review.  

The recommendation is that the Review cannot be accepted for publications in Molecules in the  present state and should  undergo  major  revisions.

Minor points.

  1. The sentence appears twice in the manuscript on P1L32 and P1L 39. Please precise.

 “. Free radical species and/or ROS induced by water radiolysis reach to a target molecule through chain reactions, or travel to a target molecule.”

  1. P2 L80 Close section or cross section? Please check.
  2. P3L125 F-FMISO “… may be recognized almost like as a standard”. In the future one may or may not use FMISO as a standard – what are the standard… . Could you please rephrase in a way that FMISO is used for hypoxia but standardization would be required or - something like that.
  3. On some Figures captions references are provided while on other references are missing. Please check.
  4. For BOLD MRI, several review papers … . Please provide the references.
  5. “uses” or use ?

P14L329 . contrast agents in MRI has been examined in early 1980’s . Could you please check if more recent data is available ? [1] 

Author Response

We provide our response to the reviewer’s comments in the attached file.

Reviewer 2 Report

The review surveys important topics. 

Specific comments:

1) The Review title does not reflect the content of the work: It should be about theranostics and MRI, but in the abstract, we can see PET, EPR imaging, etc.

Also, «In this review, we describe detection and analysis of biological information using magnetic resonance-based imaging techniques for achieving theranostic radiation therapy including recent developments of multimodal imaging techniques using redox-responsive MRI contrast agents.»

But in the review, I can see the sections like Imaging Hypoxia by PET, EPR Oxygen Mapping. It is not MRI. We can see in the Review much more information about biological processes imaging. And it is very good. Also, I haven’t seen there a lot of information about cancer theranostics. It looks like this review most of all about MRI, a bit PET examples. And only the examples for hypoxia PET imaging.

Moreover, it has fresh examples of using nitroxides for MRI of biological process. So, I think the title and abstract should be rewritten according to the information in the Review.

2) About PET. Do you want to present the examples of cancer PET imaging or only hypoxia? If you want only hypoxia imaging you have to write in the introduction, abstract, title, etc.

3) About nitroxides and MRI. I have found much more works about MRI for cancer imaging using nitroxides. According to your title, abstract I think you have to present them there:

You can google organic radical contrast agents. Some «fresh» works:

Pro-organic radical contrast agents ("pro-ORCAs") for real-time MRI of pro-drug activation in biological systems 2020

Human Serum Albumin Labelled with Sterically-Hindered Nitroxides as Potential MRI Contrast Agents 2020 from the Molecules J.

Triply Loaded Nitroxide Brush-Arm Star Polymers Enable Metal-Free Millimetric Tumor Detection by Magnetic Resonance Imaging  2018 from the Molecules J.

Synthesis, stability and relaxivity of teepo-met: An organic radical as a potential tumour targeting contrast agent for magnetic resonance imaging 2018

Nitroxide-Based Macromolecular Contrast Agents with Unprecedented Transverse Relaxivity and Stability for Magnetic Resonance Imaging of Tumors 2017

Organic Radical Contrast Agents for Magnetic Resonance Imaging 2012

Also, It is known that proteins modified with nitroxides potentially can work as OMRI agents for cancer imaging.

In vivo Overhauser-enhanced MRI of proteolytic activity

https://pubmed.ncbi.nlm.nih.gov/24729587/

And this protein conjugate possible work in the same way Human Serum Albumin Labelled with Sterically-Hindered Nitroxides as Potential MRI Contrast Agents 2020

4) Technical things Fig. 1 have to be presented more compact in two columns. Fig. 2 has to be smaller, so it will look much better, without not so worse pixels presenting. Fig. 3 B trytil radical should be presented in a common way. Pic.

«(B) Triarylmethy radical has narrow single line EPR spectrum. Oxo31 has a narrower EPR linewidth compared to that of Oxo63.»

There is the next-generation of the OX063 radical with the narrow line and in deuterated form. You can present better next-generation examples. You can see 2020-2021 years’ work of Dr. Tormyshev V.M. https://www.scopus.com/authid/detail.uri?authorId=6505790772

Novel Acetylene Derivatives of Stable Tetrathiatriarylmethyl Radicals, Reversible Dimerization of Human Serum Albumin, Methanethiosulfonate Derivative of OX063 Trityl: A Promising and Efficient Reagent for Side-Directed Spin Labeling of Proteins, In Situ Labeling and Distance Measurements of Membrane Proteins in E. coli Using Finland and OX063 Trityl Labels

5) Paper that can be suitable for the review

“Redox Imaging” to Distinguish Cells with Different Proliferative Indexes: Superoxide, Hydroperoxides, and Their Ratio as Potential Biomarkers

https://www.hindawi.com/journals/omcl/2019/6373685/

Nitroxides as Antioxidants and Anticancer Drugs https://www.mdpi.com/1422-0067/18/11/2490

Author Response

(The authors gave the same response as above.)

Round 2

Reviewer 1 Report

The  manuscript has been significanly improved . 

For now  detected  just a  problem with the  Positronium  "atom" . 

L153 to 166  Positronium is a hydrogen-like atom consisting of a positron and an electron. ..

Definitely not an atom. It is a system – oversimplified a sort of “dancing ballet” between  and e+ and e- , that  just results  in the  annihilation  process being  delayed ( compared to positron life time). The paragraph should be checked carefully.

The  time  is  one  thing  but   how do the  authors  think the 3D location is obtained ? The registration process ( for positronium) should be  from  known 3D  coordinates ? A single  camera/detector is not enough ( just register the  effect )  .

The usual collection will target only  simultaneous 511keV events and  discriminate  the  others . One needs statistical average to “calculate/approximate” the  location  ( or  additional  detector for  the  triple  annihilation processes if the  "source" allows it) .

And the 22Na  is nowadays a used for solid-state sandwich type or Doppler measurements. 22Na half life is  ~1.6 years  if correctly remember  not suitable for  the patients. . Please consider editing. 

The  references are  now  quite sound. 

I am not going to  complain about the  English  language  but please would/could  James B. Mitchell  have a  look at the text one more time, as some  passages/sentences may eventually be  improved. 

The  overall  recommendation is  for minor revisions. 

Author Response

(The authors gave the same response as above.)

Reviewer 2 Report

Thank you for the new version of the paper. See below some technical suggestions for paper revision.

1) You should avoid using abbreviations in the abstract part. One another option to present their description for non-common BOLD, TOLD.

2)  The text "Nitroxyl radical labeled amino-acid type contrast agents for achieving simultaneous drug delivery and tumor imaging has been reported [133, 134]" I think have to better rewritten "as protein [133] and amino-acid type [134]"

However, some things were overlooked.

3) Fig. 1 has to be presented more compact in two columns way.

4) Fig. 2 has to be smaller in one line way, so it will look much better, without not so worse pixels presenting.

Author Response

(The authors gave the same response as above.)
